## Original Research Article

microtubule nucleation; microtubule persistence length; plant cortical microtubules; protoxylem; stochastic simulation.

**Corresponding author:**
Eva E. Deinum
Email: eva.deinum@wur.nl

**Associate Editor:** Sebastian Wolf

# Microtubule flexibility, microtubule-based nucleation and ROP pattern co-alignment enhance protoxylem microtubule patterning

Bas Jacobs[1], Marco Saltini[1] , Jaap Molenaar[1], Laura Filion[2] and Eva E. Deinum[1]

[1] Mathematical and Statistical Methods (Biometris), Plant Science Group, Wageningen University & Research, Wageningen, 6708 PB, The Netherlands; [2] Soft Condensed Matter and Biophysics Group, Debye Institute for Nanomaterials Science, Utrecht University, Utrecht, 3584 CC, The Netherlands

## Abstract

The development of the water transporting xylem tissue in plants involves an intricate interplay of Rho-of-Plants (ROP) proteins and cortical microtubules to generate highly functional secondary cell wall patterns, such as the ringed or spiral patterns in early-developing protoxylem. We study the requirements of protoxylem microtubule band formation with simulations in CorticalSim, extended to include finite microtubule persistence length and a novel algorithm for microtubule-based nucleation. We find that microtubule flexibility facilitates pattern formation for all realistic degrees of mismatch between array and pattern orientation. At the same time, flexibility leads to more density loss, both from collisions and the microtubule-hostile gap regions, making it harder to maintain microtubule bands. Microtubule-dependent nucleation helps to counteract this effect by gradually shifting nucleation from the gap regions to the bands as microtubules disappear from the gaps. Our results reveal mechanisms that can result in robust protoxylem band formation.

## 1. Background

To meet the water demand of their above ground organs, plants depend on their water transporting tissue, the xylem. Xylem consists of an interconnected tubular network that stretches from young root tips to the above ground water sinks. Inside the xylem, the water pressure is often negative (Brown, 2013), meaning that the constituting elements have to be sufficiently strong to prevent collapse (Venturas et al., 2017). This is extra challenging close to the root tips. The earliest maturing xylem vessels, called protoxylem, mature while the surrounding tissue is still elongating (Růžička et al., 2015). Vessel elements could not stretch along if they have homogeneously thick cell walls. Therefore, protoxylem elements show banded or spiralled secondary cell wall reinforcements. These bands provide strength (Roumeli et al., 2020), but also allow for elongation. In contrast, the later maturing metaxylem has much more solid secondary cell wall reinforcements, with a number of ellipsoid gaps (Turner et al., 2007). More general, these two xylem types are studied as a model system for pattern formation in cell wall structure (Xu et al., 2022).

The patterned secondary cell wall reinforcements initially consist mostly of cellulose microfibrils deposited by cellulose synthase (CESA) complexes (Kamon & Ohtani, 2021; Turner et al., 2007). Upon maturation, these reinforcements are additionally lignified (Barros et al., 2015). CESA delivery to the membrane and their subsequent movement through it is guided by the cortical microtubule array (Chan & Coen, 2020; Crowell et al., 2009; Gutierrez et al., 2009; Paredez et al., 2006; Watanabe et al., 2015), with the cellulose microfibrils in the cell wall themselves as a secondary directing mechanism in absence of microtubule contact (Chan & Coen, 2020). Understanding how the characteristic protoxylem cell wall patterns form, therefore, requires understanding how the corresponding patterns arise in the cortical microtubule array.

Microtubule dynamics during xylem patterning can be studied by ectopic expression of the transcriptional master regulators vascular NAC (VND) 6 and 7, which induce metaxylem and protoxylem like patterns, respectively (Yamaguchi et al., 2010, 2011). In these systems,

the cortical microtubule array adopts the pattern that the subsequent secondary cell wall reinforcements will follow (Higa et al., 2024; Schneider et al., 2021). This pattern is formed in interaction with Rho-of-Plants (ROP) proteins and their downstream effectors. In metaxylem, active AtROP11 accumulates in future gap regions, where it recruits microtubule depletion domain 1 (MIDD1) and Kinesin-13A, leading to local microtubule depolymerisation (Oda et al., 2010; Oda & Fukuda, 2012, 2013). In protoxylem, a similar interaction between ROPs and microtubules is highly likely, and striated ROP and MIDD1 patterns have been observed (Brembu & Winge, 2005; Higa et al., 2024). Many ROPs are expressed in the zone of protoxylem patterning (*Arabidopsis*: only AtROP6, 9 and 10 are consistently *not* expressed) (Brady et al., 2007; Li et al., 2016; Wendrich et al., 2020). Not surprisingly, the ROPs responsible for the protoxylem pattern remain elusive as, e.g., *atrop7/8/11* triple knockouts lose the metaxylem pattern, but still form a banded protoxylem pattern (Higa et al., 2024), possibly due to a high degree of redundancy. Microtubules themselves also influence the shape of the ROP pattern by anisotropically restricting active ROP diffusion (Oda & Fukuda, 2012; Sugiyama et al., 2017), which can orient the ROP pattern along the microtubules (Jacobs et al., 2020). This coupling results in a degree of co-alignment between the orientation of the original microtubule array and the banded ROP pattern, particularly in highly aligned arrays. Previous simulations suggest that this co-alignment increases the speed of pattern formation (Schneider et al., 2021), but it has not been thoroughly quantified to what degree co-alignment is required in any previous simulation work. Consequently, we do not know the likelihood that individual arrays meet this requirement in practice.

Individual microtubules are highly dynamic, particularly at their plus-end (Fig 1a). They display phases of growth and rapid shrinkage, with stochastic switches between the two, called catastrophe and rescue (Aher & Akhmanova, 2018; Desai & Mitchison, 1997; Gudimchuk & McIntosh, 2021). As the cortical microtubules are attached to the inside of the cell membrane, they are bound to interact via frequent collisions. The outcome of these collisions depends on the relative angle of the colliding and obstructing microtubule (Dixit & Cyr, 2004, see also Fig. 1b). For small angles, the colliding microtubule bundles with the obstructing one, while for large angles it either crosses over or undergoes an induced catastrophe (Dixit & Cyr, 2004). After a crossover (Fig. 1B), the latest arriving microtubule, i.e., on the cytoplasmic side, is most likely to be severed by katanin at a later time (Fig. 1c) (Lindeboom et al., 2013; Zhang et al., 2013). Computer simulations and theoretical models have been indispensable in understanding how the above interactions can lead to the spontaneous self-organization of the cortical array into highly aligned patterns, without the aid of other patterning proteins, such as ROPs (Allard et al., 2010; Deinum et al., 2011, 2017; Deinum & Mulder, 2013, 2018; Durand-Smet et al., 2020; Eren et al., 2010; Mirabet et al., 2018; Tindemans et al., 2010). As computer simulations are more easily amended to complex situations, they have been the approach of choice in studying protoxylem development, where the ROP-specified banded pattern adds an extra layer of complexity (Jacobs et al., 2022; Schneider et al., 2021). As modelling ROPs and microtubules simultaneously remains a computational challenge, the strategy of choice has been to model them one at a time, i.e., modelling ROP patterning with a fixed aligned microtubule array (Jacobs et al., 2020) and modelling microtubule patterning with a fixed banded ROP pattern (Schneider et al., 2021). In the latter case, however, full reproduction of the patterning process with

interacting microtubules turned out to be a hard problem and has not been achieved yet (Schneider et al., 2021).

One major reason is related to microtubule nucleation. Most microtubules in the cortical array are nucleated from existing microtubules with a specific distribution of relative nucleation angles (Chan et al., 2009). The most common implementation of microtubule based nucleation (Allard et al., 2010; Deinum et al., 2011; Schneider et al., 2021), however, introduces a global competition for nucleations, i.e., regions with more microtubules attract more nucleation complexes and therefore gain even more microtubules in a positive feedback loop. This feedback loop invariably leads to highly inhomogeneous arrays (Jacobs et al., 2022) and aggregation of microtubule density in one or few bands of the protoxylem pattern only (Jacobs et al., 2022; Schneider et al., 2021). Recently, it has been shown that this so-called 'inhomogeneity problem' was a consequence of an incomplete understanding of the nucleation process (Jacobs et al., 2022). A more realistic nucleation algorithm based on detailed experimental observations in which competition occurs locally but not globally is indeed sufficient to produce regular banded protoxylem arrays in a simplified context of transverse, non-interacting microtubules (Jacobs et al., 2022). The question is whether the adoption of a more realistic nucleation algorithm (Saltini & Deinum, 2024) alone will be sufficient to produce timely band formation in the full interacting microtubule array.

Another potentially important aspect of microtubules that is often disregarded in modelling studies is their flexibility (Fig. 1g). Most cortical array studies model microtubule segments in between collision points as perfectly straight line (Chakrabortty et al., 2018; Deinum et al., 2017; Eren et al., 2010; Lindeboom et al., 2013; Schneider et al., 2021; Tindemans et al., 2010), justified by a millimetre range persistence length of isolated microtubules (Hawkins et al., 2010; Sasaki et al., 2023). In microscopic images of plant cells, however, microtubules appear less straight, e.g., (Fu et al., 2005; Nakamura et al., 2018; Schneider et al., 2017, 2021), suggesting that, in the cell context, their growth between interactions can be better described as semiflexible polymers with a sub-millimetre effective persistence length. Such a lower effective persistence length could be caused by the crowded environment of the cortex, e.g., by forces acting on microtubules generated by active cellulose synthase complexes (Liu et al., 2016) and cytoplasmic streaming (Sainsbury et al., 2008; Shaw et al., 2003). Unfortunately, the effective persistence length in plant cells has never been quantified yet, and the recent studies that do model microtubules as semiflexible polymers (Durand-Smet et al., 2020; Mirabet et al., 2018), base themselves on measurements from animal cells (Brangwynne et al., 2007; Pallavicini et al., 2014). The value of 26 $\mu$m they use seems rather low, as it is similar to the values found *in vitro* for high density interacting microtubules with reduced persistence length in the presence of MAP70-5, addition of which introduces small circular microtubule bundles (Sasaki et al., 2023). Actual persistence length is, therefore, likely higher in transverse arrays and protoxylem bands. The effect of different degrees of flexibility on protoxylem patterning remains an open question. Flexibility may both help and hinder the patterning process, as incorrectly oriented microtubules may find their way back to a band, while correctly oriented ones may curve out of it.

Here, we study the effect of three different concepts on protoxylem microtubule patterning: (1) the degree of co-alignment between microtubules and the microtubule-hostile future gap regions likely specified by ROPs, (2) realistic microtubule-based nucleation through a recently developed, computationally efficient

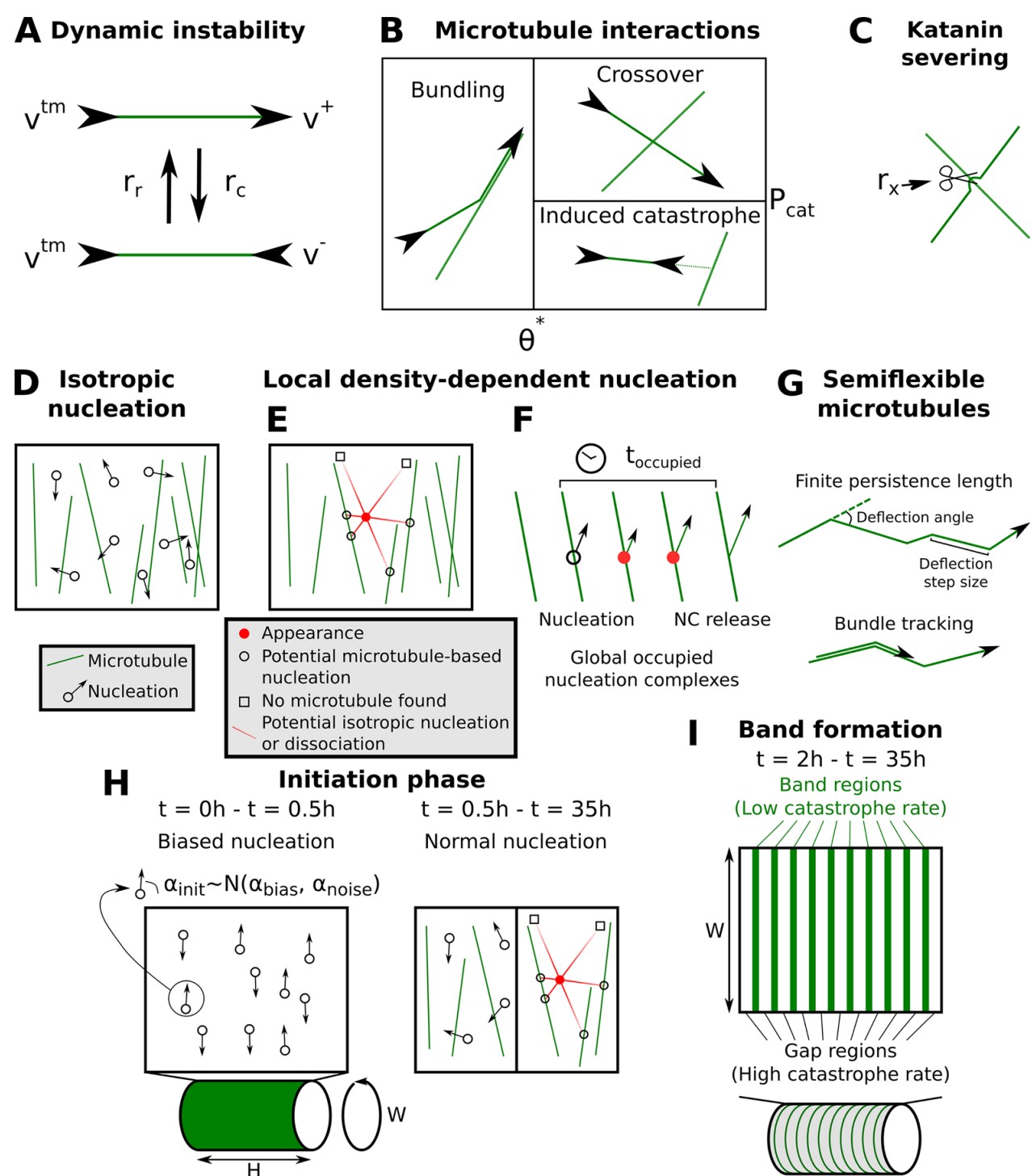

**Figure 1.** Microtubule simulation details. (a) Microtubule plus ends grow or shrink at constant rates $v^+$ or $v^-$, respectively, and minus ends retract at constant rate $v^{tm}$. Spontaneous catastrophes (switch from growing to shrinking) occur at a rate $r_c$ and rescues (switch from shrinking to growing) at a rate $r_r$. (b) Microtubule-microtubule collision outcomes depend on the collision angle. At angles below, $\theta^*$ the impinging microtubule bundles with, i.e., continues growing along, the obstructing one. At greater collision angles, the incoming microtubule undergoes an induced catastrophe with a probability $P_{cat}$ and crosses over the other microtubule otherwise (Tindemans et al., 2010, 2014). (c) Katanin severs the overlying microtubule at crossovers at a constant rate $r_x$ per crossover. (d) With isotropic nucleation, new microtubules appear at a constant rate $r_n$, in a uniformly random location and direction. (e) With local density-dependent nucleation, new microtubules appear in a way that more realistically mimics the behaviour of nucleation complexes (NCs), taking into account their appearance at the membrane, diffusive movement and potential microtubule binding. (f) In addition, nucleation complexes stay occupied for a duration $t_{occupied}$, temporarily reducing the global nucleation rate. (g) For simulations with semiflexible microtubules, a finite persistence length is achieved via discrete deflections in the microtubule growth direction. In addition, microtubules in bundles follow their bundle around bends below an angle $\theta_b$ ('bundle tracking'). Deflection angles in cartoons are exaggerated for visibility. (h) In band formation simulations, an initial transverse array is artificially enforced by drawing nucleation angles in the first half hour of simulated time ($\alpha_{init}$) from a normal distribution with an average of $\alpha_{bias}$ and a standard deviation of $\alpha_{noise}$. (i) Protoxylem band formation is simulated with predefined band and gap regions, where the catastrophe rate in the gap regions is increased by a factor $f_{cat}$ after a $2h$ initiation phase, following (Schneider et al., 2021), except that $f_{cat}$ is reduced to 3.

nucleation algorithm that captures the critical aspect of localised positive feedback (Saltini & Deinum, 2024), and (3) a realistic degree of microtubule flexibility.

## 2. Methods

### 2.1. Microtubule simulations

We performed our simulations using an extended version of the cortical microtubule simulation software CorticalSim (Tindemans et al., 2014), fast, event-driven software for simulating cortical microtubule dynamics and interactions on the cell surface (cortex) (Chakrabortty et al., 2017; Deinum et al., 2011, 2017; Schneider et al., 2021; Tindemans et al., 2010, 2014). Compared to previous simulations on protoxylem (Schneider et al., 2021), we included the newly developed nucleation algorithm from Saltini and Deinum (2024) and extended the simulation software to allow for microtubule flexibility. Unless stated otherwise, we used a cylindrical geometry with dimensions representative of the VND7 cells used in the experiments from which we get our data (a height of 60 $\mu m$ and a radius of 7.5 $\mu m$). For a detailed overview of all parameter values, see Supplementary text A. Compared to (Schneider et al., 2021), the induced catastrophe probability $P_{cat}$ has been reduced to 0.09 and severing at crossovers happens by default.

### 2.2. Microtubule dynamics

The microtubules are modelled as connected series of line segments that together have one plus end and a minus end. The minus end retracts at a constant speed $v^{tm}$ (Fig. 1a) simplifying observed minus end dynamics (Shaw et al., 2003), while the plus end mimics dynamic instability by switching between growing and shrinking states (catastrophes and rescues, Fig. 1a). When a microtubule collides with another microtubule, this results in a bundling event for collision angles $\theta < \theta^*$ after which the microtubule continues growing along the obstructing microtubule, and an induced catastrophe or crossover event for larger angles (Fig. 1b). Additionally, when a microtubule collides with the edge of the bounding cylinder, it also undergoes a catastrophe, which helps favour a transverse array orientation (Ambrose et al., 2011). Finally, any microtubule crossing over another can undergo a severing event at the intersection with a rate $r_x$ per crossover, creating a new shrinking plus end and retracting minus end (Fig. 1c).

### 2.3. Microtubule flexibility

In order to model the underlying flexibility of the microtubules, we extended the CorticalSim model drawing inspiration from the method in (Mirabet et al., 2018). Specifically, we introduced 'deflection' events that abruptly change the microtubule growth direction. For a full description, see Supplementary text B. In summary: the deflection step size, i.e., the length a microtubule grows straight before the next deflection occurs, is drawn from an exponential distribution with mean $\bar{l}$. The deflection angle is drawn uniformly from $[-m, m]$, where $m$ is the maximum deflection angle calculated to obtain the desired persistence length given $\bar{l}$. Note that deflection angles with absolute values lower than a minimal deflection angle $q = 0.1°$ were set to zero to avoid numerical artefacts. With these parameters we can control the persistence length $l_p$ as shown in Supplementary text C. To prevent microtubules from leaving bundles at every deflection point, we made microtubules follow their bundle along bends smaller than 10° (Fig. 1g).

In our simulations, microtubule bundles consist of multiple microtubules on the same trajectory, without any space in between. Therefore, there is no distinction between microtubules on the edges of the bundle, which could in principle deflect outwards, and microtubules on the inside, that do not have any room to deflect at all. To approximately overcome this limitation, we reject a fraction $n/(n+1)$ of deflections in bundles, where $n$ is the number of other microtubules in the bundle at the point of the deflection. Additionally, in bundles, we would expect most microtubules to stay with the bundles through small bends. To incorporate this feature, we force microtubules to track their bundles along bends, as long as the bending angle is not too large. Specifically, since bundling events create bend points and these may have large angles (up to 40°), we implemented a maximum bundle tracking angle of $\theta_b = 10°$. If a bundle splits with an angle below this value, an incoming microtubule randomly follows one of the bundles, proportional to their occupancy.

### 2.4. Microtubule nucleation

In many microtubule modelling studies (Deinum & Mulder, 2018; Durand-Smet et al., 2020; Mirabet et al., 2018; Tindemans et al., 2010) new microtubules appear by isotropic nucleation, i.e, with random, uniformly distributed locations and orientations (Fig. 1d). However, it has previously been shown that in practice, most microtubules are nucleated from existing microtubules, with a specific distribution of angles (Chan et al., 2009). In our model, we explicitly include this nucleation mechanism, modelling the angle distribution following the approach of Deinum et al. (2011). Additionally, we perform some simulations with purely isotropic nucleation for comparison.

To avoid the inhomogeneity problem of previous microtubule-bound nucleation algorithms that distribute bound nucleations proportional to the microtubule density [like (Deinum et al., 2011)], we use a recently developed, more realistic nucleation algorithm dubbed 'local density-dependent nucleation' (Saltini & Deinum, 2024) that effectively approximates the diffusion of nucleation complexes from their appearance at the plasma membrane to the point where they either dissociate or nucleate. In this approach, nucleation complexes are not modelled as explicitly diffusing particles with additional dynamics, but handled implicitly via instantaneous appearance events, occurring at a rate $r_{ins}$. During such an event, the algorithm draws an 'appearance point' for a complex at a random, uniformly distributed position within the simulation domain. Then, three outcomes are possible: i) the complex dissociates without nucleation, ii) a microtubule-bound nucleation, or iii) an unbound nucleation.

From the appearance point, $n = 6$ possible linear paths (or metatrajectories) are generated (Fig. 1e). We set $n = 6$ to reasonably consider the possible directions where nucleation complexes can diffuse. These paths are oriented in $n$ equally spaced directions, with an overall offset that is chosen at random. Potential sites for microtubule-bound nucleation correspond to points where one of the paths intersects a microtubule at a distance $d$. The probability that a nucleation complex reaches a site at this distance from its appearance point without nucleating before is:

$$p_{\text{bound}}(d_i) = e^{\frac{-r_u d_i^2}{4D}}, \tag{1}$$

where $r_u$ represents the rate of unbound nucleation, and $D$ is the nucleation complex diffusion coefficient at the membrane.

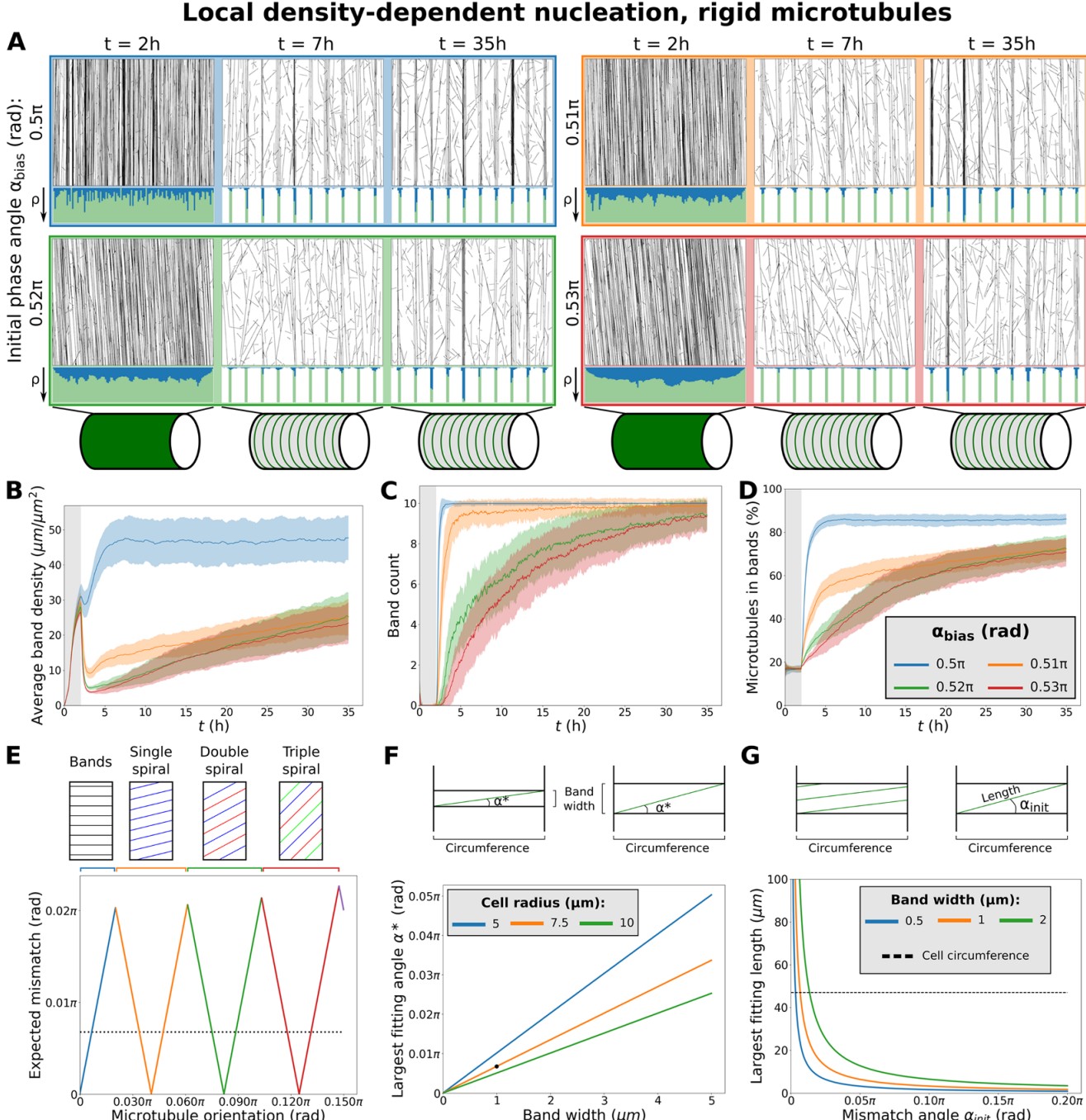

**Figure 2.** Fast protoxylem patterning is sensitively dependent on co-alignment between microtubules and the underlying pattern. (a) Snapshots from protoxylem simulations using starting arrays with bias angles $\alpha_{bias}$ of $0.5\pi$ (90°), $0.51\pi$ (91.8°), $0.52\pi$ (93.6°) and $0.53\pi$ (95.4°) in the first half hour with only minor deviations ($\alpha_{noise} = 0.032\pi$ rad). Histograms below showing local microtubule density $\rho$ share the same axis within a time series. For the different time series, the $\rho$-axis ranges from 0 to 125, 96, 89 and 77 $\mu m/\mu m^2$, respectively. (b) Average microtubule density in the band regions. (c) Number of populated bands, defined as bands with a microtubule density greater than three times the average density in the gaps. (d) Percentage of the total microtubule length residing in the bands. Quantities in (b–d) were calculated from 100 simulations. The band formation phase starts at $t = 2h$, i.e., at the end of the grey area. Lines indicate the average and shaded areas the standard deviation. (e) Minimal expected mismatch angle of a spiral (or banded) ROP pattern following a microtubule array of different orientations based on geometrical constraints for a distance of 6 $\mu m$ between the centres of bands and a cylindrical domain with a radius of 7.5 $\mu m$. Dotted line indicates the largest mismatch angle for which a microtubule can still span the cell's circumference within a band. (f) Largest mismatch angle $\alpha^*$ at which a microtubule (bundle) can fit within a band along the entire circumference of the cell. Black dot indicates default simulation values. (g) Largest length that a microtubule (bundle) can have while still fitting entirely within a band region at varying mismatch angles. Band width in simulations is 1 $\mu m$ (orange line). Dashed line indicates cell circumference for comparison.

For computational efficiency, we only consider intersections at distances less than $d_{\max} = 1.5$ $\mu m$, beyond which we consider $p_{\text{bound}}(d)$ to be zero based on our parameter choices. The probability of a potential microtubule-bound nucleation event is therefore given by the sum of $p_{\text{bound}}(d)$ over all directions where an intersection was found. Conversely, the probability of a potential isotropic nucleation event is

$$p_{\text{iso}} = 1 - \frac{1}{n}\sum_j p_{\text{bound}}(d_j), \qquad (2)$$

with the summation over *j* including only those paths that intersect a microtubule. We execute a single nucleation event either isotropic or on the first microtubule intersected by metatrajectory *i* based on these probabilities.

Based on empirical observation (Jacobs et al., 2022; Nakamura et al., 2010), a complex reaching the lattice of a microtubule dissociates with probability 76%, while a freely diffusing complex dissociates with probability 98% (see (Jacobs et al., 2022)). Hence, before committing to either a microtubule-based or isotropic nucleation event, we reject the appearance event with a probability of 76% or 98%, respectively. These uniform post-hoc rejection probabilities are justified by the observation that membrane dissociation rates are very similar for free and microtubule-bound nucleation complexes that do not nucleate (Jacobs et al., 2022).

Isotropic nucleations are executed at the original appearance point. In case of a microtubule-based nucleation, a new microtubule is nucleated parallel, antiparallel, or branched to either side with an angular distribution with the mode at 35° with respect to the parent microtubule, exactly as in (Deinum et al., 2011), based on the experimental data by (Chan et al., 2009).

Previous observations showed that nucleation complexes move from gap regions to band regions as bands start emerging, with the total number of complexes remaining relatively constant (Schneider et al., 2021). To accommodate this observation, we model an overall fixed number of nucleation complexes $N_{tot}$, which can be either free or occupied. Upon successful nucleation, a nucleation complex becomes occupied for a duration $t_{occupied} = 60s$ (Fig. 1f), which is the average time until a nucleation complex is released from a new microtubule by katanin (Nakamura et al., 2010). When handling an appearance event, the attempted nucleation is immediately rejected with a probability equal to the fraction of currently occupied complexes.

### 2.5. Protoxylem simulations

We modelled the local activity of proteins specifying the banded pattern (most likely ROPs and their downstream effectors (Brembu & Winge, 2005; Higa et al., 2024; Oda & Fukuda, 2012)), using a difference in catastrophe rate between predefined band and gap regions as in (Schneider et al., 2021). Experimental observations of protoxylem development show that microtubule patterning starts from a well-established transversely oriented array (Schneider et al., 2017, 2021). We, therefore, started our simulations with a two-part initiation phase similar to (Schneider et al., 2021). The first 30 minutes, all nucleations occurred at random positions with a variable angle $\alpha_{init}$, drawn from a normal distribution with a mean of $\alpha_{bias}$ and a standard deviation of $\alpha_{noise}$ (Fig. 1h), to firmly establish array orientation. This was followed by 90 minutes of the nucleation mode for the remainder of the simulation, to generate a more realistic array microstructure. After the two hour initiation phase, we subdivided the simulations domain into ten 1 $\mu$m wide band regions separated by 5 $\mu$m wide gap regions to simulate protoxylem band formation. We increased the catastrophe rate in the gap regions by a factor $f_{cat}$ (Fig. 1i), whereas the parameters in the band regions remained unchanged, similar to simulations by Schneider et al. (Schneider et al., 2021).

Preliminary simulations using a sinusoidal profile on the catastrophe rate (Klooster, 2017) show that discrete bands can also form with a more gradual profile. A gradual increase of the catastrophe rate in the gaps would correspond more closely to the gradual increase of MIDD1 speckles close to the edges of the gaps observed in (Higa et al., 2024). Our simple band-gap profile, however, is

consistent with our previous simulation studies, and, moreover, easier to parametrize, as it corresponds to the way the experimental data in (Schneider et al., 2021) is quantified.

### 2.6. Expected mismatch angles

Although a ROP pattern is expected to follow the general orientation of the initial microtubule array, this match may not be exact, as the ROP pattern needs to wrap smoothly around the geometry, while maintaining an intrinsic band spacing (Jacobs et al., 2020). We have previously shown (Jacobs et al., 2020) that the orientation of a spiral ROP pattern that maintains the distance between bands follows:

$$\vartheta = \arcsin\left(\frac{H \cdot n}{W \cdot n_{bands}}\right), \tag{3}$$

where $H$ is the domain length, $W$ the domain circumference, $n$ the spiral number (1 for a single spiral, 2 for a double spiral, etc.), and $n_{bands}$ is the number of bands in an equivalently spaced banded array. For our 10 bands, this equation gives a set of discrete angles that a ROP pattern is likely to follow. Assuming that the ROP pattern will always adopt the orientation closest to that of the microtubule array (reasonable at least for low spiral numbers; see (Jacobs et al., 2020)), microtubule arrays with orientations in between these discrete spiral angles will have the mismatch shown in Fig. 2e.

### 3. Results

### 3.1. Strong co-alignment of the microtubule array with a pre-existing band pattern facilitates rapid microtubule band formation

Some degree of co-alignment between the initial (aligned, but still homogeneous) microtubule array and the developing ROP pattern is biologically realistic, since the orientation of the microtubule array helps shape the orientation of the ROP pattern (Jacobs et al., 2020; Oda & Fukuda, 2012). However, the orientation of a protoxylem ROP pattern is also influenced by geometrical constraints, as it has to form either rings or spirals that wrap around cell's the circumference (Jacobs et al., 2020). As such, the orientation of the microtubule array can vary continuously, whereas the ROP pattern can only follow in discrete jumps. Depending on the orientation of the microtubule array, we calculated that the resulting mismatch between the orientation of the ROP pattern and microtubule array can be as high as 3.6° (Fig. 2e).

Therefore, we tested the sensitivity of microtubule patterning to this mismatch angle using simulations with different starting array orientations. When starting with a microtubule array that is strongly co-aligned with the orientation of the band regions, we found that microtubule bands could form rapidly, both for isotropic nucleation (Fig. S.3 and S.4) and for local density-dependent nucleation (Fig. 2a–d), due to density-loss in gaps (Fig. S.5). A slight mismatch of 3.6° (0.02π rad) in this co-alignment, however, already resulted in an extremely slow band formation process for both nucleation modes, similar to simulations in (Schneider et al., 2021) with isotropic nucleations.

We observed that with a mismatch up to 1.8° (0.01π rad), the microtubule density in bands was maintained, whereas with mismatches of at least 3.6°, this density was lost at the beginning of the patterning process after which bands were 'rediscovered' at a slow rate (Fig. 2b). Simulations with a smaller (2 $\mu$m) cell radius

more representative of endogenous protoxylem rather than VND7-induced hypocotyl cells yielded similar results (Fig. S.6). These results suggest that co-alignment between the initial microtubule array and the underlying pattern proposed to be formed by ROPs is an important ingredient for timely microtubule band formation. However, the co-alignment we can expect from the ROP pattern orienting after the microtubule array (Fig. 2e) is currently insufficient for band formation without full breakdown and rediscovery.

As a proxy for calculating the maximum tolerable mismatch angle, we calculated the largest mismatch angle for which a straight microtubule could still span the circumference once while staying within the band (Fig. 2f) as well as the largest stretch of microtubule that could fit in a band given the mismatch angle (Fig. 2g). The point at which band formation began to suffer from the mismatch was similar to the point at which a single microtubule (bundle) could no longer stay in a band region along the entire circumference of the cell (Fig. 2f). For greater mismatch angles the largest length of microtubule (bundle) that could fit within a band without bending rapidly decreased (Fig. 2g). Variation of cell circumference and band width within the biologically relevant range suggests that the co-alignment requirement for straight microtubules is too strict to guarantee timely band formation.

The effect of co-alignment alone, therefore, did not explain timely band formation, and the use of more realistic microtubule nucleations did not improve upon this effect.

### 3.2. Microtubule flexibility can lead to density loss from bands under isotropic nucleation

Previous models with microtubule flexibility (Durand-Smet et al., 2020; Mirabet et al., 2018) used values of $l_p$ = 20–30 $\mu m$ measured *in vivo* in animal cells (Brangwynne et al., 2007; Pallavicini et al., 2014). For microtubule dynamics parameters based largely on measurements in developing protoxylem (Schneider et al., 2021), these persistence length values caused so many extra collisions that it resulted in a loss of density and alignment even in simulations without band formation (Fig. S.7a–c). Possibly, these persistence length values are too low for plant cells. Sasaki et al. (2023) found average persistence lengths of 60 $\mu m$ and 100 $\mu m$ in their *in vitro* gliding assays with high and low microtubule densities, respectively. Measurements of microtubules suspended in flow cells even gave persistence lengths of around 2 $mm$ for non-interacting microtubules (Sasaki et al., 2023). Since the deflections due to microtubule interactions are modelled directly in our simulations, it is not just the 'intrinsic' persistence length that contributes to the measured persistence length and so the measured persistence length will be smaller than the value we should use to control deflections. Therefore, we investigated multiple larger persistence lengths, up to the millimetre range consistently measured in *in vitro* experiments with individual microtubules (Hawkins et al., 2010; Sasaki et al., 2023). For persistence lengths of hundreds of micrometres, aligned arrays did form (Fig. S.7a–c). Curiously, for these persistence lengths, the edge-induced catastrophes were not always sufficient to give the array a transverse orientation (Fig. S.7d). For our study of protoxylem band formation we, therefore, continued using the biased initiation phase.

Band formation was actually hindered by semiflexible microtubules when using isotropic nucleation. For $l_p$ = 100 or 200 $\mu m$, stable starting arrays could be formed, but density was lost when band formation started (Fig. 3 and S.8). It would seem, therefore, that the extra flexibility, rather than helping microtubules find bands, actually makes microtubules already in bands bend out and

suffer from the increased catastrophe rate in the gap regions. Proper bands only formed for more rigid microtubules with $l_p$ = 500 or 1000 $\mu m$.

The loss of density in the band regions at the lower persistence lengths could at least partially be counteracted when a significant portion of isotropic nucleations was moved to the gap regions (Fig. S.9). Such a shift could be expected to occur dynamically in cells as gap regions start to empty, when taking into account microtubule-bound nucleations, suggesting that a more realistic implementation of nucleations could be essential to band formation when taking microtubule flexibility into account.

### 3.3. Local density-dependent microtubule-based nucleation helps to keep bands populated with semiflexible microtubules, even for misaligned starting arrays

When combining semiflexible microtubules with local density-dependent nucleation, band formation improved for $l_p$ = 100 $\mu m$, but most microtubule density was still initially lost, and took a long time to recover. However, band formation was now possible at $l_p$ = 200 $\mu m$, lower than for isotropic nucleation, as nucleations were automatically allocated to the denser band regions as gap density started to decrease (Fig. 3b–e).

Furthermore, the combination of semiflexible microtubules and local density-dependent nucleation also greatly improved timely band formation for a significant mismatch between the orientations of the starting array and the band pattern. A mismatch as high as 18° (0.1$\pi$ rad) in the angle of the nucleations in the initiation phase still yielded a partially banded pattern after five hours of band formation (Fig. 4a–d). Similar results were obtained for a smaller cell radius of 2 $\mu m$ (Fig. S.10). As measured in Fig. 4, the use of edge-induced catastrophes reduced the mismatch of the starting array compared to the bias angle. Still, the actual mismatch corresponding to the 18° intended mismatch ($\alpha_{bias}$ = 0.6$\pi$ rad) was about 10.8° (0.06$\pi$ rad), substantially more than the worst mismatches we expected theoretically for the default cell radius (Fig. 2).

For persistence lengths in the millimetre range, band formation improved at strong co-alignment, but slowed band formation at weaker co-alignment (Fig. S.11 and S.12). This effect was still not nearly as strong as for rigid microtubules (Fig. 2), suggesting that timely band formation can be achieved for a wide range of possible microtubule persistence lengths.

A reason for the increased tolerance for mismatches might be that the semiflexible microtubules inherently cover more different angles, of which a substantial portion could align with the band regions, even though the average orientation does not. This would give the array more opportunities to correct its course, by micro-tubules bending back into the array, or by 'wrong' orientations getting catastrophes in gap regions and rescues at a point where the orientation better matches the assumed underlying ROP pattern. An investigation of the angles at the start of the band formation process in simulations with microtubule flexibility showed that there is indeed a broad distribution of microtubule segment angles (Fig. 4e).

## 4. Discussion

We have identified three important ingredients for microtubule patterning in developing protoxylem by doing extensive simulations: (1) sufficient co-alignment between the microtubule array and the underlying ROP pattern, (2) microtubule flexibility,

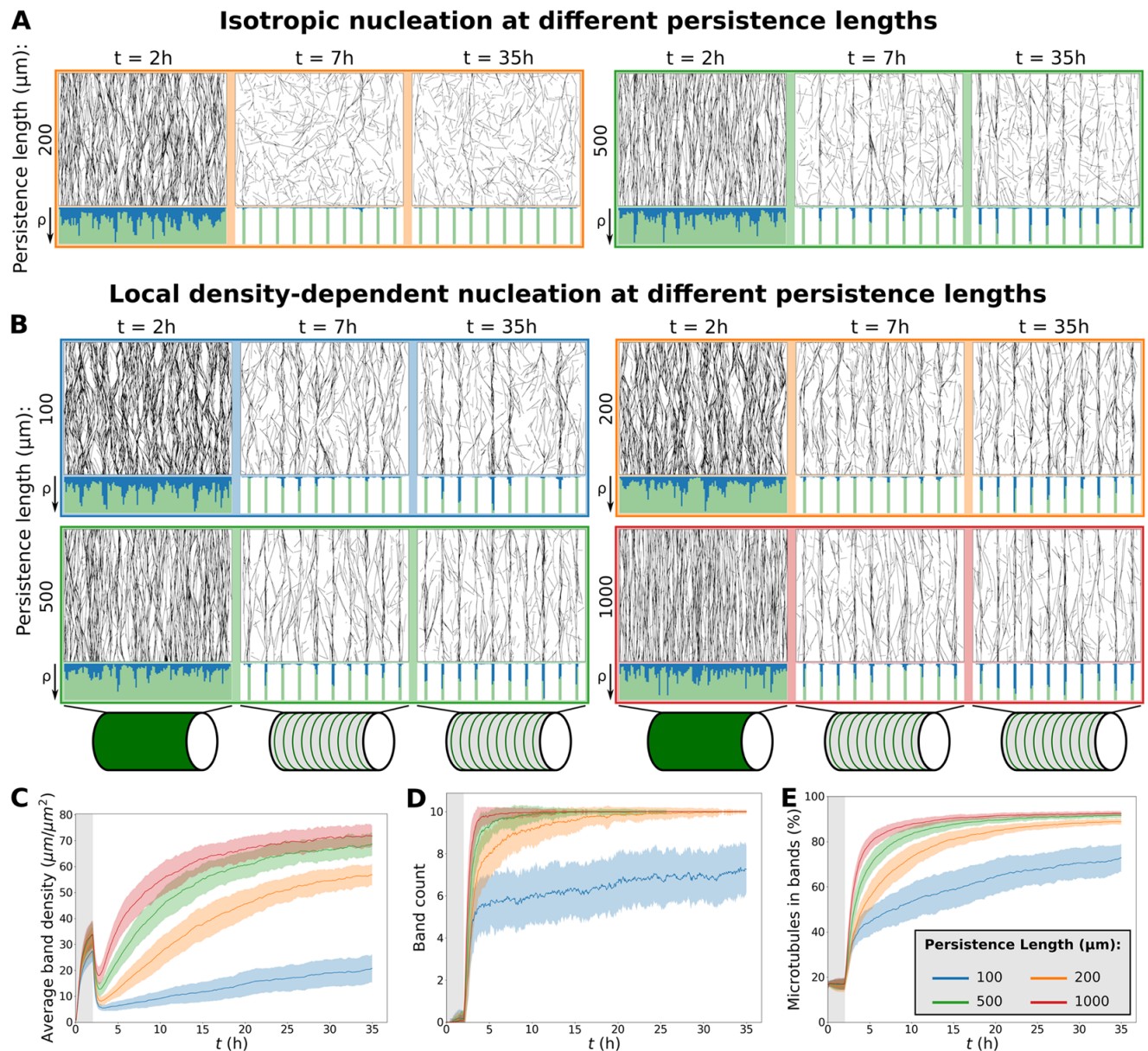

**Figure 3.** At lower persistence lengths, too much density is lost to maintain bands. This problem is less severe with local density-dependent nucleation than with isotropic nucleation. (a and b) Snapshots from protoxylem simulations with isotropic (a) and local density-dependent nucleation (b) for different microtubule persistence lengths. Starting arrays were obtained with transverse nucleations in the first half hour ($\alpha_{bias} = 0.5\pi$, $\alpha_{noise} = 0.032\pi$ rad). Histograms below showing local microtubule density $\rho$ are plotted on the same scale within a time series. For the different time series, the $\rho$-axis ranges from 0 to 53, 63, 74, 124, 150 and 126 $\mu m/\mu m^2$, respectively. (c) Average microtubule density in the band regions. (d) Number of populated bands, defined as bands with a microtubule density greater than three times the average density in the gaps. (e) Percentage of the total microtubule length residing in the bands. Quantities in (c–e) were calculated from 100 simulations with local density-dependent nucleation. The band formation phase starts at $t = 2h$, i.e., at the end of the grey area. Lines indicate the average and shaded areas the standard deviation.

and (3) realistic microtubule-based nucleation. Together, these aspects allow microtubule bands to form on a realistic time scale in fully interacting microtubule array simulations with idealised microtubule dynamics parameter values based on measured data (Schneider et al., 2021).

One important aspect that we did not model explicitly is the interplay between microtubule patterning and ROP patterning. Here, we modelled the ROP pattern as a static, banded prepattern, whereas in reality, it is most likely shaped by a reaction diffusion process in which a combination of positive feedback activation and a difference in diffusion between active and inactive ROPs leads to spontaneous pattern formation (we extensively reviewed the details of such processes in plants elsewhere (Deinum

& Jacobs 2024)). This ROP patterning has been modelled for both metaxylem (Nagashima et al., 2018) and protoxylem (Jacobs et al., 2020). For protoxylem, the formation of the banded ROP pattern involves microtubules acting as ROP diffusion barriers that orient the pattern (Jacobs et al., 2020). We partly modelled this orienting effect by generating starting arrays with various degrees of co-alignment. However, while real microtubule arrays generally align in a transverse orientation before the start of protoxylem patterning, they rarely have exactly the same orientation along their entire length (Schneider et al., 2017, 2021). The resulting bands also tend to have a variation in their orientation and they are not all perfectly straight or equally spaced in ways that reflect initial densities and orientations in the starting arrays (Schneider et al., 2017, 2021).

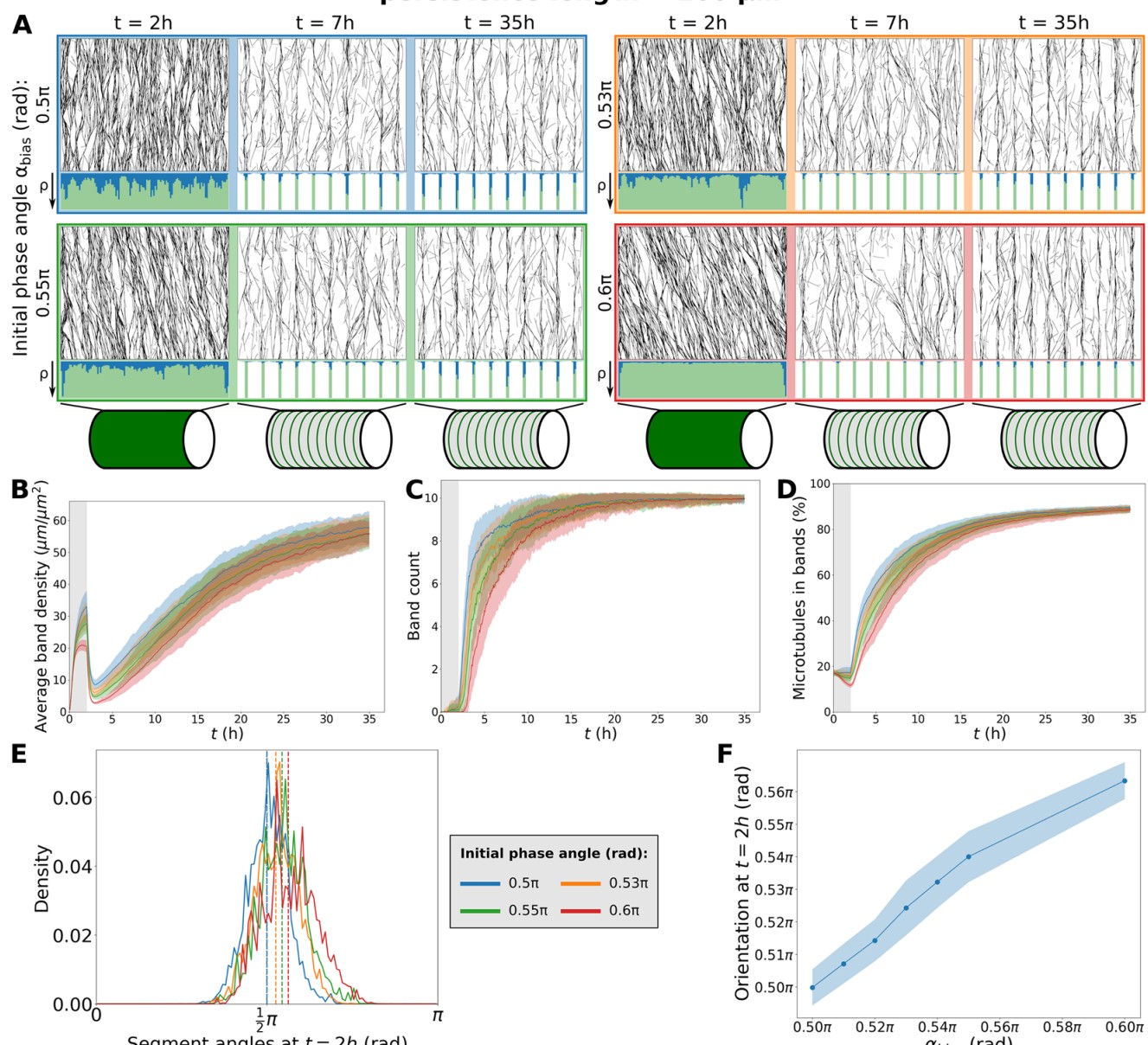

**Figure 4.** With local density-dependent nucleation (Fig. 1), greater mismatches in the initial orientation still allow fast band formation. (a) Snapshots from protoxylem simulations for $l_p = 200\ \mu m$ with local density-dependent nucleation using starting arrays with different bias angles $\alpha_{bias}$ in the first half hour with only minor deviations ($\alpha_{noise} = 0.032\pi$ rad). Histograms below showing local microtubule density $\rho$ are plotted on the same scale within a time series. For the different time series, the $\rho$-axis ranges from 0 to 98, 204, 146 and 471 $\mu m/\mu m^2$, respectively. (b) Average microtubule density in the band regions. (c) Number of populated bands, defined as bands with a microtubule density greater than three times the average density in the gaps. (d) Percentage of the total microtubule length residing in the bands. (e) Distribution of microtubule segment angles, weighted by segment length, at $t = 2h$ from the individual example simulations shown in (a). Dashed lines indicate the the overall array orientation. (f) Average array orientation at $t = 2h$ as a function of the bias angle in the initiation phase. Quantities in (b–d), and (j) were calculated from 100 simulations. The band formation phase starts at $t = 2h$, i.e., at the end of the grey area. Lines indicate the average and shaded areas the standard deviation.

These observations suggest that the real underlying ROP pattern also provides co-alignment at a local level, which may help microtubule band formation, given some flexibility to follow the curved features. At the same time, the possibility of curved bands results in an increased rate of straight-growing microtubules leaving band regions, which we found could be detrimental to band formation. To help growing microtubules follow these curved bands, microtubule associated proteins (MAPs) involved in microtubule bundling may be important, e.g., MAP65-8, which is expressed in developing xylem (Kubo et al., 2005). In addition, microtubules must be sufficiently flexible to follow these curves.

Our simulations demonstrated the importance of microtubule flexibility even when assuming straight band and gap regions. Flexibility improved the ability of microtubules to follow predefined band regions in spite of small mismatches in the orientation of the microtubules and the band regions. However, our simulations also indicated a trade-off, where increased flexibility means more microtubules curve into gap regions where they are more likely to

suffer catastrophes. Microtubule associated proteins may, however, reduce this effect in practice by preventing microtubules from bending into the gap regions.

The need for microtubule flexibility becomes even more obvious when we consider metaxylem patterning, where arrays need to form circular or ellipsoid gaps. In microscopic pictures, microtubules appear to curve around these gaps (Sasaki et al., 2017; Stöckle et al., 2022). The gapped structure also means that microtubule patterning cannot rely on co-alignment with the ROP pattern as in protoxylem patterning. Therefore, metaxylem patterning may require additional proteins to help the microtubules form this structure that are absent or less important in protoxylem patterning. Recent evidence indicated the involvement of microtubule-associated protein MAP70-5, which lowers microtubule persistence lengths, enabling the formation of microtubule loops (Sasaki et al., 2023; Stöckle et al., 2022). This is consistent with earlier observations that MAP70-5 lines the borders of the pits and its overexpression increases the ratio of pitted to spiral cell wall patterns (Pesquet et al., 2010). MAP70-5 has also been observed in protoxylem (Sasaki et al., 2023), where less extreme microtubule curvature is required, suggesting a more subtle role in modulating persistence length. Cortical (CORD) proteins may also be involved in metaxylem patterning. These proteins disorder microtubules by partially detaching them from the membrane (Sasaki et al., 2017), possibly facilitating corrections in the microtubule orientation. Other potential factors include 'Boundary of ROP domain 1' (BDR1), Wallin, and actin networks that form in the pits, though these may only act in a later stage, during formation of the pit borders (Sugiyama et al., 2019).

In addition to the importance of microtubule flexibility, our simulations also showed the importance of realistic microtubule-based nucleation. We have previously shown the importance of locally saturating microtubule-based nucleation for array homogeneity in Jacobs et al. (2022). There, we assumed a constant, uniform supply of nucleation complexes as an important source of local saturation. Here, we relaxed this assumption by allowing microtubules to draw from a global pool of nucleation complexes, while maintaining a local density-dependence of the nucleation rate. Consequently, the reduction of microtubule density in the gap regions would free up nucleation complexes that could then boost the nucleation rate in the bands. This effect helped compensate for microtubule loss from catastrophes suffered by microtubules leaving the band region. This partial shift in the location of nucleation complexes is in line with microscopic observations of nucleation complexes during protoxylem development (Schneider et al., 2021) and upon oryzalin treatment (Jacobs et al., 2022).

Our results also hint at an effect of the cell size, which is relevant when translating results from models and experiments on VND7 cells to the narrower cells naturally developing into protoxylem. With narrower cells, the mismatch angle between the initial microtubule array and the ROP pattern can be larger, as the transitions from bands to single spirals to double spirals lead to larger changes in orientation. With rigid microtubules, the band formation process was actually less tolerant of these mismatches. However, with flexible microtubules, the tolerance to mismatches was very similar for different cell sizes.

Another interesting finding was that the introduction of a finite persistence length made it harder to obtain global alignment from edge-induced catastrophes alone. This reduction of global alignment compared to local alignment has been observed before with rigid microtubules on large domains (Deinum, 2013), but microtubule flexibility reduced the domain size at which locally aligned

patches formed. Microtubule flexibility, therefore, increases the importance of local orienting cues, such as the observed sensitivity of microtubules to mechanical stress (Colin et al., 2020; Eng et al., 2021; Hamant et al., 2008; Hamant et al., 2019; Hoermayer et al., 2024; Sampathkumar et al., 2014; Takatani et al., 2020; Verger et al., 2018). It has also been suggested that, depending on the spacing of microtubule-membrane linkers, individual microtubules may also tend towards a longitudinal orientation to minimize their bending energy before anchoring (Bachmann et al., 2019; Tian et al., 2023). It remains to be seen, however, whether this mechanical force would be strong enough to overrule the orienting forces arising from collective microtubule interactions (Saltini & Deinum, 2024).

The lowest persistence lengths reported in Sasaki et al. (2023) ($\approx 30\ \mu m$) in high density assays including MAP70-5, are comparable to the 20–30 $\mu m$ measured in animal cells (Brangwynne et al., 2007; Pallavicini et al., 2014). The largest persistence length in gliding assays in Sasaki et al. (2023), $\approx 100\ \mu m$ in low density assays without MAP70-5, was on the low side for timely band formation, but there remains a large gap with the millimetre range values they observed in flow cells. This difference suggests a large uncertainty in the value of the microtubule persistence length in plant cells, as well as a possibility for cells to modulate it. In both gliding assays, Sasaki et al. (2023) found a 2–3 fold reduction of persistence length by the addition of MAP70-5. In absence of sufficient membrane attachment, strong microtubule bending can be induced by forces generated by active cellulose synthase complexes (Liu et al., 2016) and cytoplasmic streaming (Sainsbury et al., 2008; Shaw et al., 2003). Therefore, good candidates for persistence length modulation are proteins involved in microtubule-membrane linkage, such as CELLULOSE-MICROTUBULE UNCOUPLING (CMU) proteins (Liu et al., 2016), and certain IQ67-Domain (IQD) proteins (Bürstenbinder et al., 2017), of which IQD13 functions in metaxylem development (Sugiyama et al., 2017). Loss of CMU proteins even results in increased lateral movement of the microtubule lattice, though with an average displacement of around 100 nm, this effect is not likely to have a strong impact (Liu et al., 2016). IQD proteins can also be regulated dynamically, in particular through calcium signalling, as they have calmodulin-binding domains (Bürstenbinder et al., 2017; Kölling et al., 2018). This kind of regulation may dynamically influence microtubule persistence length. Therefore, persistence lengths measured in one situation may not necessarily apply to the next, making it necessary to obtain *in vivo* persistence length measurements for plant cells of different types.

Our results suggest that the quantification of the relevant *in vivo* persistence length is important for understanding many aspects of cortical array behaviour, but its measurement will not be easy. Various methods of obtaining persistence lengths from experimental observations exist, for example, (1) quantifying how fast the orientation decorrelates along the microtubule, (2) comparing the direct distance between any two points on the microtubule with the contour length between them, (3) comparing the direct distance between any two points on the microtubule to the distance between the microtubule and the midpoint of the line segment joining these two points, and (4) estimating the curvature distribution of the microtubules (Lamour et al., 2014; Wisanpitayakorn et al., 2022). For accurately determining the persistence length using any of these methods, however, it is important that the data consists of sufficiently long microtubule stretches. Since we independently model bundling, we need the persistence length of non-interacting microtubules, which poses a problem for performing such experimental measurements *in vivo*, as the distance between

microtubule-microtubule interactions typically is several orders of magnitude smaller than expected values of $l_p$. Theoretically, the measurement could be further complicated by CESA complexes introducing lateral displacement of the microtubule lattice from the position in which it originally polymerised (Liu et al., 2016). These effects, however, are likely only relevant in specific mutants, like the *cmu1cmu2* mutant (Liu et al., 2016). With future advances in high-throughput experiments and their automated analysis, this fundamental challenge may be tackled, at least in part, by using large data sets derived from relatively sparse cortical arrays.

Both experimental and theoretical studies show that cortical microtubules and ROPs mutually affect each other, and hence, the patterns formed by the interacting system (Jacobs et al., 2020, 2022; Oda & Fukuda, 2012; Schneider et al., 2021; Sugiyama et al., 2017). From ROP simulations with implicit microtubules, we expect that the speed at which the microtubule array adapts to the current ROP pattern has a substantial impact on the ROP patterns formed (Jacobs et al., 2020). Microtubule-based nucleation can substantially increase the lifetime of microtubule bundles (Saltini & Deinum, 2024), and hence, likely has a big impact on this speed. Also microtubule flexibility could impact this speed, via tuning the impact of curved features in the ROP pattern on the local microtubule array. With ROPs typically modelled using continuum (partial differential equation) descriptions [e.g., (Jacobs et al., 2020), (Nagashima et al., 2018), for xylem], and microtubules as discrete stochastic entities, integrating both systems in a single simulation environment requires the development of an efficient interface between formalisms, which would then also enable integration with other cues affecting microtubule behaviour like wall mechanical stresses (Hamant et al., 2008). Our results on the importance of microtubule flexibility, microtubule-based nucleation and pattern co-alignment will help interpret the results of such integrated simulations. The first two factors will likely have a substantial quantitative, if not qualitative, impact on the patterning outcomes of the interacting system, and are thus important factors to consider when further studying (proto)xylem patterning.

## 5. Conclusion

In conclusion, we have shown that co-alignment, microtubule flexibility, and local density-dependent nucleation are important aspects of protoxylem patterning. Our findings lay the groundwork for future studies on patterns generated by microtubule-ROP interactions. These studies may include simulations of other systems, such as metaxylem, as well as simulations that combine existing ROP models (e.g., (Jacobs et al., 2020)) with microtubule simulations.

**Funding statement.** The work of M.S. was supported by a Research Grant from HFSP (Ref.-No: RGP0036/2021) to E.E.D.

**Competing interests.** The authors declare none.

**Author contributions.** B.J., M.S., J.M., L.F. and E.E.D. designed the research. B.J., M.S. and E.E.D. performed the research. B.J. analysed data. B.J., M.S. and E.E.D. wrote the paper. B.J., M.S., J.M., L.F. and E.E.D. reviewed and edited the paper.

**Data availability statement.** We are in the process of producing a new release of CorticalSim, which will be made public along with the publication of (Saltini & Deinum, 2024). The release will become accessible via (Tindemans & Deinum, 2017).

**Supplementary material.** The supplementary material for this article can be found at https://doi.org/10.1017/qpb.2024.17.

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
