## [Reviewer Report]

The manuscript “Protoxylem microtubule patterning requires ROP pattern co-alignment, realistic microtubule-based nucleation, and moderate microtubule flexibility” is part of a series of publications from Bas Jacobs, member of the Deinum lab in Wageningen, in which very carefully performed simulations are used to trace the role of microtubules and their dynamic spatio-temporal regulation in generating the spiral/band patterns of the protoxylem secondary cell wall. The authors explain very well why the flexibility of microtubules is an important but still under-researched topic. The manuscript is written in a very clear and understandable way. The illustrations are also of great didactic value despite the complicated parameter landscape the authors have to introduce. The main findings of the manuscript are:

(1) pre-alignment of the MT array prior to patterning enables rapid MT band formation

(2) MTs that are too flexible (less persistent) hinder rapid MT band formation, and

(3) local density-dependent MT nucleation can compensate for the delays caused by incorrect/incomplete pre-alignment of the MT array prior to patterning.

The authors embed those in the current state of research, particularly with regard to the experimental results. However, the authors fail to discuss suitable key experiments (or model parameters that should be quantified [how?] to improve the simulations) that could be taken up as impulses for experimental plant biologists. I see this as a good opportunity to even clearer justify the classification of this manuscript into the journal “Quantitative Plant Biology”. I suggest accepting the manuscript as soon as the above & below comments have been addressed.

Minor issues:

- the last sentence in the abstract (“Our results reveal the main mechanism ...”) seems too strong, considering that this manuscript does not provide additional experimental evidence for the microtubule persistence lengths/array misalignment angles assumed in the simulations and that it is only “a” model. I suggest softening the tone of this statement.

- in the sentence “Upon maturing, they ...”, it is not clear what “they” refers to. Please improve.

- the authors should highlight how microtubule persistence lengths are measured experimentally/in-silico. I’m sure it’s written in their text, but I wasn’t immediately made aware of it while reading.

- in the last paragraph of the introduction, at the sentence “Here, we study the effect of ...” point (1) needs amendment. It should read “microtubule-hostile future gap regions”, not “band regions”.

Major comments/questions:

- the authors adapt their simulations to an experimental framework (VND7-inducible hypocotyl cells) already established in the paper by Schneider/Klooster et al (Nat Commun, 2021). Could the authors comment on what impact smaller cell sizes would have? Would the predicted phenomena also be observable in the much smaller cells of the endogenous xylem?

- the authors should briefly discuss whether MAP70-5, which apparently reduces MT persistence length (i.e. makes MTs more flexible), is necessary in the protoxylem at all. After all, the ring/spiral bands have much larger radii than the metaxylem pits.

---

## [Reviewer Report]

In the submitted manuscript by Bas et al. the authors use computational modelling, to simulate microtubule (MT) band formation in developing protoxylem cells. The authors have a good record in such approaches and recently gave indications that ROP band pattern formation in protoxylem cells depends on diffusion restriction by MTs. This ROP pattern is counter-localizing the observed MT pattern in this cell type.

In this Manuscript the authors show that multiple properties of MTs are required for efficient band formation of MTs in protoxylem cells. They simulate effects of different co-alignment angles with an underlying ROP pattern, ranges of persistent length and flexibility in MT elongation angles, and different types of nucleation patterns.

The manuscript is well written, the methods are explained in detail and the results are illustrated nicely. The theoretical model provided by the authors can provide a good resource to test different parameters and properties of MT and ROP regulating proteins in protoxylem cells. However, I a few major and minor points that the authors should address before publication.

Major:

1: Persistent length of MTs: The authors mention that in plants no proper measurements of MT persistence length were performed. To strengthen the outcome of the proposed model, it would be great to measure the actual persistence length of MTs in protophloem cells, especially before band formation. This would show if the model is based on realistic values or if other parameters need to be considered.

2: The authors should include lateral MT displacement effects in their model, or at least discuss the potential effects of such displacement on band formation and MT clustering. The displacement range of 50-100 nm as shown by Liu et al (https://doi.org/10.1016/j.devcel.2016.06.032) is small but could be another crucial factor in combination with MT flexibility.

3: At multiple points of the manuscript, the authors mention the underlying potential ROP pattern that is required for band formation. The authors should show this pattern that is predetermined in the initiation phase.

4: The most crucial point in this manuscript, especially considering the publication of ROP pattern formation by this group (https://doi.org/10.1016/j.jtbi.2020.110351), is that the initiation phase is not addressed in more detail. The authors assume an underlying ROP pattern that MTs align to. However, in the mentioned previous publication it is shown that the establishment of this ROP pattern required MTs as diffusion barriers which requires an already existing stronger density of MTs in a band patter. This is a classical chicken-egg problem, for which we lack experimental evidence. This should be phrased more clearly.

Very likely MT band formation and ROP accumulation in gaps is established simultaneously and requires negative feedback between both, without a clear preexisting pattern. It would be great if the authors would provide simulations with a more flexible ROP pattern in the initiation phase or in the following simulation. For example, the authors could assume a more dotted ROP activity pattern, or a generally more diffuse ROP activation rather than a clear banded pattern. This co-formation of both patterns could lead lead to synergistic effects and a faster pattern establishment.

Minors:

1: “Many ROPs are expressed in the zone of protoxylem patterning [22]”. Also refer to later publications using single cell transcriptomics and a potential better resolution especially in developing protoxylem cells. Additionally name the expressed ROPs.

2: “but it has not been thoroughly quantified to what degree of co-alignment is required.” Something in this sentence is not right. I guess its just deletion of the “of”.

3: MT density in Figure 2D, 3E, 4D (and supplements).

A) The authors should also provide the density of MTs in GAPs as raw values to better show if there is a decrease in these regions. Otherwise, it is also possible that the provided GAP/Band-Ratio is dominated by an increasing density in Bands without decrease in the GAPs.

B) I suggest changing the Ratio in the density to Band/GAP to have an increasing value, this is more consistent and more intuitive in relation the neighboring panels.

---

## [Reviewer Report]

Review of Jacobs et al. for Quantitative Plant Biology

Summary

-------

In this manuscript, Jacobs et al. describe recent work on a model exploring the dynamics of the cortical microtubule (MT) array in protoxylem patterning. Through the interaction of ROP and MTs, both components end up forming a banded pattern on the cell membrane that promotes the formation of banding in the cell wall. At an abstract level, this system represents a “mutual-inhibition” pattern-forming interaction that is quite different from superficially similar and well-studied pattern-forming purely reaction-diffusion systems, making it of potential interest to both biological and mathematical audiences. Extending previous work, here the authors fix the membrane ROP distribution into regularly spaced parallel bands along the circumference of a cylindrical plant cell and study factors that influence the evolution of the cortical MT array into complementary bands. They study the influence of the orientation of the initial MT array, the flexibility of the MTs and the details of MT nucleation on the timing and extent of MT band formation.

General issues

--------------

I found it hard to read this paper without also reading the authors' previous work, especially refs [16] and [24]. The authors should include more about what was and what was not done in their previous work and how this paper addresses some of the prior model shortcomings. Also, the manuscript refers to [16] for details of the extension of the earlier model predating that work but I was not able to find the “Methods and Supplementary Note 1” which is referred to in [16] but seemingly not included. For the sake of reproducibility, more detail is required here and/or better referencing to the correct sources.

One specific point that can be made clearer in the introduction is the nature of the coupled ROP-MT system. Most of the description makes it sound like the two-way feedback between ROP and MTs is necessary for protoxylem patterning but then it states that this coupling “speeds up” pattern formation as if the coupling isn’t necessary for patterning. This distinction is important to reconcile because the present study fixes ROP in bands and considers only the dynamics of MTs on the fixed ROP background which I understand to be a “divide and conquer” approach to modelling the coupled system (fix one side of the coupling at a time and study the dynamics of the other). What is not made clear in the introduction is that previous work from the group has done a simplified version of the opposite (fixed an idealized oriented MT array and simulated ROP spatiotemporal dynamics) [24] and also a previous version of “fixed ROP - dynamic MTs” [16]. It would make it easier for the reader to appreciate the value of the present study if the intro covered this history, making the assumptions from previous work more transparent and clarifying how the present work fits into that history (e.g. making it clear what was and was not done in [16] and what is done here and how those all fit together). It would also be nice to see comments in the discussion summarizing what remains to be done and perhaps some reference to why it hasn’t yet been done (I assume a fully coupled model requires more modelling work or possibly more experimental knowledge to be filled in).

Regarding figures, I am not sure if there is an expectation from the journal limiting the number of figures but Fig 2,3,4 could be made easier to digest if each were split into one showing the MT visualization and a second one with the time courses and other quantifications.

Specific issues

---------------

--Title--

The claim that all three factors are required should really be that they are sufficient and even then, without a dynamic ROP model, that isn’t obvious.

“...requires ROP pattern co-alignment” - although I believe this to be a true fact, the study treats ROP as fixed and the necessity of ROP patterning is an assumption rather than a conclusion of this study and suggests more than the study shows.

Claiming that moderate persistence length is required is undermined by Fig 3 which shows that MT bands form for all Lp tested except perhaps 100 um.

--Methods--

Microtubule flexibility:

Persistence length is usually used to describe the stiffness of a MT where one is thinking of an elastic rod with a certain flexural rigidity subject to thermal forcing while suspended in a fluid. When non-thermal forces influence the shape, the persistence length can be shorter, as found in the papers referenced by the authors. However, on the cortex in plant cells, it’s not clear that existing estimates of persistence length are relevant. For a cortically anchored MT in a plant cell, its shape is determined by fluctuations of the free tip that are frozen by the attachment of an anchor somewhere along that free tip. A very high anchoring rate (implicitly or explicitly assumed in most models, in the form of having MTs follow geodesics) would essentially prevent any fluctuations leading to an arbitrarily high persistence length. Very rare anchoring might allow the thermally driven shapes to produce estimates closer to the values measured in suspension. Additional forces (like cytoplasmic streaming) could bring the measured persistence length down. Without direct measurement of Lp in plants, I don’t see any reason to adopt a sub-mm figure. I like that the authors explored a range of persistence lengths and I am not troubled that short persistence lengths discouraged banding but I think the framing of this part of the paper ignores the subtlety of the difference between persistence lengths in different contexts (e.g. suspended in fluid versus embedded in an actin meshwork versus cortically anchored).

Also, the authors use p for persistence length but earlier used subscripted p for various probabilities. Perhaps L<sub>p</sub> or l<sub>p</sub> which are often used might remove ambiguity.

Inhomogeneity problem:

Although there is a paragraph talking about the “inhomogeneity problem”, it was difficult to make sense of what the actual problem was until looking over previous work from the authors. As the present study is described as improving on what the authors' previous model did, describing that would be very helpful. Some mention of the positive feedback loop created by nucleation factors binding directly to randomly chosen segments of existing MTs (instead of regions of membrane first, as in the new model) would be helpful.

Initial nucleation bias:

The statistics of the initial nucleation bias are described using the mean and variance. Using variance (with units of rad<sup>2</sup>) makes it hard to parse the number. Also, it is given (in the Fig 2 caption) as 0.01 rad<sup>2</sup> where all other angles are given relative to pi. It would be nicer if it was given as σ<sub>noise</sub>=0.032 π.

--Results--

Angle mismatch study (3.1):

The authors tested their model’s convergence to a banded state when an initial non-banded MT array was highly ordered and skewed relative to the fixed ROP bands. They found that for very small angle mismatches (<1.8<sup>o</sup>), convergence was quick but for slightly larger angles (eg 3.6<sup>o</sup>) convergence was slow because the pre-existing array had to disassemble and a new one reform at the correct orientation. This makes intuitive sense although the sensitivity at such small angles is a bit surprising. What is less clear is the relevance of this study. This half-model (ROP fixed, MTs dynamic) seems to be a study case whose main purpose ought to be facilitating the formulation of the full model. In my mind the first step would be to test whether the steady state of the half model is stable to perturbations that would be relevant in the context of the full model. An angular shift of a full non-banded MT array is not a small perturbation of the steady state of either the half or the full model (the initial array is also “hyper-organized” because of the nucleation bias). It certainly gives some information about how the half model behaves with certain initial conditions but I don’t see its relevance to anything else. What is the motivation for this test? A study more directly related to the general problem of formulating a full ROP-MT model should be included. Or if that is not the long-term goal here then a better motivation for this work and this particular section would be helpful.

In Fig 2, the MT density histograms are helpful to gauge band formation but they apparently have different vertical scales. It would improve their use to give some indication of scale on each one. Perhaps the axis label could be ρ on one graph and eg 1.5 ρ on another etc.

Also in Fig 2, “Average density bands” should be “Average band density”?

Microtubule flexibility study (3.2):

Where the authors use the term “biased initiation phase” (end of the first paragraph of the section), they should specify that they only use α<sub>bias</sub>=0.5π.

Nucleation / semiflexible study (3.3):

“As measured in Fig 4F, our boundary conditions reduced the mismatch ...” - If the authors are referring to edge-induced catastrophes, I suggest saying so explicitly because the term boundary condition is not used for this in the methods.

--Discussion--

The words required/requirement etc are used in a few places (discussion and elsewhere) but are generally too strong for what has been shown. The sense of requirement here must be qualified by all the other things that have been included and omitted from the model. I would prefer the word(s) be avoided altogether unless actually justified.

Regarding the claim of requirement of moderate persistence lengths, the interpretation I prefer to make from the authors' results is:

“Values of L<sub>p</sub> that are likely not relevant in plants cause problems for the model but more reasonable values are fine, for both isotropic nucleation and more realistic nucleation”.

The claimed requirement of moderate flexibility (i.e excluding both higher and lower flexibility) seems inappropriate. They do mention in section 3.2 that larger L<sub>p</sub> leads to fewer transverse arrays (without ROP bands), which is an interesting observation, but this is not clear from Fig S.5D. In that figure, the colours seem to fade with larger L<sub>p</sub> (what colour saturation denotes is not mentioned) but changes in the number of transverse arrays with L<sub>p</sub> is not obvious by eye. Furthermore, that comment in 3.2 seems to contradict a comment in the discussion: “Another interesting finding was that the introduction of a finite persistence length made it harder to obtain global alignment from edge-induced catastrophes alone”. Maybe this is referring to something else but in any case the observations being described are a bit unclear. Specifically, what problems were found with larger persistence lengths?

Typos

undergoes and induced  undergoes an induced

discreet  discrete

Fig 3 caption: “...less severe local density-dependent...”  “...less severe with local density-dependent...”

Fig 3 titles: “different persistence length”  “different persistence lengths”

“actual mismatch of corresponding”  “actual mismatch corresponding”

---

## [Editor Report]

Dear Eva Deinum.

thank you for your patience in awaiting the outcome of the revision process of your manuscript. My apologies for the unusual long time it has taken to recruit suitable reviewers. 

We have now received evaluations of three experts and as you can see from their comments, all three point out the interest and value of your study. However, they also list several request that should be addressed before publication. We hope you will find the comments of the reviewers helpful and look forward to receiving a revised version of your manuscript.

---

## [Reviewer Report]

The authors of the Manuscript “Microtubule flexibility, microtubule-based nucleation and ROP pattern co-alignment enhance protoxylem microtubule patterning.” by Jacobs et al. improved their manuscript according to the suggestions and answered the raised issues. I understand and agree with the explanations the authors provide in most points.

The only remaining concern is the connection to ROPs. For me the phrasing in the manuscript might be misleading me and potential readers. This, I think, is solely an issue of phrasing in manuscript to make it clear.

- My comment #3 was aiming to better highlight the regions in which ROPs themselves are active in Figure 1I, e.g. showing future MTs regions (bands with low catastrophe) in green, ROPs defined regions (GAPs with high catastrophe rate) in magenta. This should make it clearer in which pattern both are being or resulting in.

- The simulation of ROP patterning in previous publications needed predefined arrays to introduce ROP anisotropy, and MT band formation in this publication depend on a predefined array of higher and lower catastrophe rates. The manuscript does not clearly phrase if the authors assume an unknown mechanism that predefines the array, or if this is mainly due to the underlying ROP pattern. This is due to the wording of “underlying” ROP pattern implies that the ROP pattern is fixed and the predeterminant for the simulations. I would appreciate if the authors differentiate more clearly between an assumed (so far unknown) predefined array of differential MT catastrophe rate or if this array is the actual underlying ROP pattern.

- in section 3.1. first sentence “…microtubule array and this underlying ROP pattern…”:

“This” should be changed as it is not referring to a previous sentence anymore at this position.

Besides these minor comments I suggest the publication of this manuscript.

---

## [Reviewer Report]

The present manuscript by Jacobs et al. deals with the simulation of pattern formation of microtubule networks in developing xylem tissue. More specifically, it investigates how the flexibility (bendability or persistence length) of microtubules, their mode of nucleation, and the misalignment with an underlying ROP GTPase pattern influence the formation of microtubule bands (and gaps) in silico. While the latter two parameters have already been investigated in two previous publications (Jacobs et al., 2019 and 2020), the investigations into persistence length in particular represent an important innovation, both in experimental and simulation terms. In addition, these three parameters are cleverly combined to form a coherent simulation platform. The present work should therefore be seen in the light of this series of investigations and further narrows down the possible biophysical parameters of this system. A very helpful read for experimentalists and modelers.

The work is excellently written and begins with a figure describing the essential parameters (Fig. 1). The results section as well as the conclusion part is clearly and concisely outlined and refers to additional material (Appendix A-C) in a reasonable manner. I recommend the manuscript for publication after minor comments have been addressed.

Minor comments/questions:

(1) Higa et al. (2024) showed that MIDD1 clusters within the gap midline. To what extent could this observation change (or modify) the assumptions made here (e.g. regarding microtubule dynamics parameters in the gaps)?

(2) First paragraph of the results section (3.1): What do “initial microtubule organization” and “this underlying ROP pattern” mean here? The assumption is that a ROP pattern is formed first and then the microtubules respond to it. So, to which state does “initially” refer? Could the authors please explain in more detail why the ROP pattern “must form rings or spirals”? What then is the case with metaxylem? I think this could be stated more clearly to avoid confusion.

(3) Third paragraph of the results section (3.1): “more representative of protoxylem rather than VND7 yielded similar results”... should this be better described as “more representative of endogenous protoxylem rather than VND7-induced (or transdifferentiated) hypocotyl cells yielded similar results”?

(4) Fourth paragraph of the results section (3.1): Is it known to the authors that there are more spirals in larger cells (e.g. VND7-induced hypocotyls)? Are smaller (endogenous) protoxylem cells predominantly more band-like rather than spiral-like?

(5) Results section (3.2): The microtubule-stabilizing drug taxol increases cortical microtubule density. This should limit the influence of persistence length. Are previously published experimental observations of taxol-treated transdifferentiating cells in agreement with what has been reported here?

(6) A final, somewhat aesthetic comment: the order of referencing of figure panels is somewhat confusing (e.g. for Fig.1 in the background chapter and in appendix A-C in the methods section).

---

## [Editor Report]

Dear authors, 

as you will see, all reviewers appreciate your revisions to your manuscript and support publication. However, reviewer 1 and 2 haves some suggestions that you might consider for the final version of your manuscript.